# Identification of a Potential Founder Effect of a Novel *PDZD7* Variant Involved in Moderate-to-Severe Sensorineural Hearing Loss in Koreans

**DOI:** 10.3390/ijms20174174

**Published:** 2019-08-26

**Authors:** Sang-Yeon Lee, Jin Hee Han, Bong Jik Kim, Seung Ha Oh, Seungmin Lee, Doo-Yi Oh, Byung Yoon Choi

**Affiliations:** 1Department of Otorhinolaryngology-Head and Neck Surgery, Seoul National University Hospital, Seoul 04401, Korea; 2Department of Otorhinolaryngology-Head and Neck Surgery, Seoul National University Bundang Hospital, Seongnam 13620, Korea; 3Department of Otorhinolaryngology-Head and Neck Surgery, Chungnam National University Hospital, Daejeon 35015, Korea

**Keywords:** haplotype, founder effect, *PDZD7*, p.Arg164Trp

## Abstract

PDZD7, a PDZ domain-containing scaffold protein, is critical for the organization of Usher syndrome type 2 (USH2) interactome. Recently, biallelic PDZD7 variants have been associated with autosomal-recessive, non-syndromic hearing loss (ARNSHL). Indeed, we identified novel, likely pathogenic *PDZD7* variants based on the American College of Medical Genetics and Genomics/Association for Molecular Pathology (ACMG/AMP) guidelines from Korean families manifesting putative moderate-to-severe prelingual ARNSHL; these were c.490C>T (p.Arg164Trp), c.1669delC (p.Arg557Glyfs*13), and c.1526G>A (p.Gly509Glu), with p.Arg164Trp being a predominantly recurring variant. Given the recurring missense variant (p.Arg164Trp) from our cohort, we compared the genotyping data using six short tandem-repeat (STR) markers within or flanking *PDZD7* between four probands carrying p.Arg164Trp and 81 normal-hearing controls. We observed an identical haplotype across three out of six STR genotyping markers exclusively shared by two unrelated hearing impaired probands but not by any of the 81 normal-hearing controls, suggesting a potential founder effect. However, STR genotyping, based on six STR markers, revealed various p.Arg164Trp-linked haplotypes shared by all of the affected subjects. In conclusion, *PDZD7* can be an important causative gene for moderate to severe ARNSHL in Koreans. Moreover, at least some, if not all, p.Arg164Trp alleles in Koreans could exert a potential founder effect and arise from diverse haplotypes as a mutational hot spot.

## 1. Introduction

Usher syndrome (USH) is the leading cause of hereditary sensory deaf-blindness in humans [1]. Based on the severity and progression of sensorineural hearing loss (SNHL), age at onset of retinitis pigmentosa (RP), and presence or absence of vestibular dysfunction, USH can be classified into the following three subtypes: USH1, USH2, and USH3 [2]. Of these, USH2 is the most common type, characterized by moderate-to-severe hearing loss, preserved vestibular function, and progressive retinal photoreceptor degeneration [3]. Variants in the *USH2A*, *ADGRV1*, *WHRN*, and *PDZD7* genes are associated with USH2 in humans [4,5,6].

*PDZD7* encodes a PDZ domain-containing scaffold protein, which plays a critical role in the organization of the USH2 interactome [7]. Formation of the USH2 quaternary protein complex is based upon heterodimerization between PDZD7 and whirlin and a subsequent dynamic interplay between USH2 proteins via their multiple PDZ domains [8]. *PDZD7* was initially suggested to be a causative gene of RP, a modifier of homozygous *USH2* variants, and a contributor of digenic inheritance (along with *ADGRV1*) to the development of USH2 [6,7,8]. In a *PDZD7*-knockdown zebrafish model, an USH-like phenotype, exacerbation of retinal cell death, and reduction in ADGRV1 localization in the connecting cilium were observed [6]. Interestingly, *PDZD7* was reported to be a causative gene of DFNB57 [9,10,11,12,13,14]. The morphological and functional roles of PDZD7 in the organization of cochlear hair cells have been validated in vivo and in vitro. *PDZD7*-knockout mice elicited a loss of architecture of the stereocilia bundle at the ankle-link region, resulting in the attenuation of mechanotransduction energy [7,8]. Therefore, PDZD7 is a fundamental component of normal auditory function.

To the best of our knowledge, all reported subjects with biallelic variants of *PDZD7* demonstrate autosomal-recessive, non-syndromic hearing loss (ARNSHL) without a visual phenotype, whereas monoallelic variants of *PDZD7* act as genetic modifiers of USH2. In other words, the identification of biallelic recessive variants in *PDZD7* seems to be a pathognomonic sign of ARNSHL [9,10,11,12,13,14], suggesting that diverse phenotypes could be related to *PDZD7* variants and the presence of possible genotype–phenotype correlations. However, ARNSHL in association with biallelic *PDZD7* variants has only been reported in 10 families to date in the literature [10,11,12,13,14,15], mostly without longer-term follow-ups for hearing loss and RP. This demonstrates the need to characterize the phenotypic spectrum of biallelic *PDZD7* variants among a larger cohort.

Herein, we report three novel variants of *PDZD7* from four unrelated Korean families with a segregated moderate-to-severe ARSNHL. Notably, we identified a recurring missense variant (p.Arg164Trp) of *PDZD7*; therefore, we sought to elucidate whether this variant (p.Arg164Trp) exerts a founder effect or is a mutational hot spot. By short tandem-repeat (STR) genotyping, we evaluated whether p.Arg164Trp is linked to a single common haplotype with a founder effect. Our study would potentially have a clinical impact with respect to the incorporation of p.Arg164Trp in molecular genetic testing and understanding auditory phenotypes.

## 2. Results

### 2.1. Clinical Phenotyping

In the SB394 family at the initial visit, the two affected subjects (SB394–761 and SB394–762) carrying the pathogenic PDZD7 variants were a 4-year-old girl and a 1-year-old boy. The proband (SB394–761) showed a bilaterally symmetrical severe non-syndromic SNHL, indicating an average threshold of 70 dB hearing level based on play audiograms (Figure 1A). The proband’s sibling (SB394–762) demonstrated bilaterally symmetrical moderate-to-severe non-syndromic SNHL, based on auditory brainstem response threshold (ABRT) and auditory steady-state response (ASSR) data. There was no change in the hearing ability of the affected subjects over a 6-month follow-up period. In the temporal bone computed tomography (TBCT) and internal auditory canal magnetic resonance imaging (IAC MRI), brain parenchymal lesions and inner ear anomalies, such as enlarged vestibular aqueduct syndrome (EVAS), were not observed in the affected subjects. The proband, SB394–761, has used hearing aids and received speech therapy, but recently underwent simultaneous bilateral cochlear implantation due to delayed language development (receptive and expressive language scores less than 1%, based on preoperative speech evaluation). During the operation, a slim straight electrode, CI 522 (Cochlear Company), was inserted into the cochlea, to preserve the presumed low-frequency hearing. None of them in the SNUBH394 family exhibited signs of nystagmus and balancing disorder. Furthermore, age-appropriate ophthalmologic examinations, including slit and fundus examinations, revealed no definite abnormalities in the proband.

In the SB297 family, the affected subject, SB297–597, who was 6 months old at the time of examination, was referred due to failure of the newborn screening test. There was no response to distortion product otoacoustic emission (DPOAE) on both ears. She had symmetrical moderate-to-severe non-syndromic SNHL in both ears, according to the ABRT and ASSR results (Figure 1B). SB297–597 showed a response to 55 dB click sounds on both ears at 6 and 12 months. No inner ear anomalies were found by IAC MRI analysis. She was not able to undergo ophthalmologic examinations due to young age, necessitating a follow-up electroretinogram (ERG) for differentiating the existence of the retinal disease.

In the SH246 family, the affected subject (SH246–577) had symmetrical moderate-to-severe non-syndromic SNHL in both ears (Figure 1C). DPOAE analysis demonstrated no responses in either ear. SH246–577 showed a response to 55 dB click sounds in both ears at 20 months. ASSR testing exhibited an average threshold of 60 in the right ear and 65 dB in the left ear for SH246–577. Their parents showed normal hearing in both ears.

In the SB78 family, the affected subject, SB78–137, had symmetrical moderate non-syndromic SNHL in both ears (Figure 1D). ABRT testing showed a response to 50 dB click sounds in both ears at 12 months. According to play audiograms, the average hearing threshold demonstrated 50 dB hearing level in both ears at 20 months.

Overall, all affected subjects demonstrated symmetrical moderate to severe non-syndromic SNHL in both ears, based on age-appropriate audiological examinations. Although there was no hearing deterioration during the follow-up period, for these subjects, longer-term follow-up is necessary due to the absence of eye phenotype.

### 2.2. Variant Identification and Classification

We identified four unrelated families (SB394, SB297, SB246, and SB78) carrying *PDZD7* variants as a most likely genetic etiology. The number of candidate variants found in the four families was determined based on our filtering strategy of the Exome sequencing data (Figure 2). Whenever possible, segregation analysis was completed.

Three novel recessive *PDZD7* variants were detected in these four families with moderate-to-severe prelingual ARNSHL. We observed compound heterozygosity for a frameshift (c.1669delC: p.A557fs) and a missense variant (c.490C>T: p.Arg164Trp) in family SB394, compound heterozygosity for two missense variants (c.1526G>A: p.G509E and c.490C>T: p.Arg164Trp) in family SH246, and homozygosity for a recurring missense variant (c.490C>T: p.Arg164Trp) in families SB297 and SB78 (Figure 3).

Interestingly, p.Arg164Trp variant of *PDZD7*, which was detected from all four families, demonstrated a relatively high detection frequency of 0.003205/11 in the Korean Reference Genome Database (KRGDB). This may suggest that p.Arg164Trp could be an incidentally detected, non-pathogenic SNP. However, the following observations suggest otherwise. The c.490C>T (p.Arg164Trp) variant of *PDZD7* was consistently predicted as ‘damaging’ via SIFT (Sorting Intolerant from Tolerant) and PolyPhen-2 in silico analyses. The frequency from global minor allele (ExAC; 0.00005/6 and GnomAD;0.00004775/12) was much lower than that from KRGDB. Furthermore, the PDZD7 residues of p.R164 were well conserved among the orthologs of several species (Appendix A), despite the low GERP (Genomic Evolutionary Rate Profiling) + + score of –1.62. Finally, this allele was detected in a trans configuration with a pathogenic truncation variant, c.1669delC: p.Arg557Glyfs, and was always detected either as compound heterozygosity or as homozygosity in affected subjects, strongly suggesting the pathogenicity of p.Arg164Trp. Therefore, p.Arg164Trp allele appeared to emerge as a predominant and recurring novel allele among Korean moderate-to-severe ARSNHL. According to the American College of Medical Genetics and Genomics/Association for Molecular Pathology (ACMG/AMP) guidelines [16,17], the p.Arg164Trp was detected as a homozygote from two probands and as a compound heterozygote in a trans configuration with a pathogenic truncation variant (p.Arg557Glyfs*13), thereby providing two points compatible with ‘strong evidence’ regarding PM3 rules related to hearing loss. The p.R164W was rare enough to satisfy rarity in population databases, leading to PM2. Additionally, PP1 from SB394 with one relative also segregating this allele could be applied, but PP3 (Rare Exome Variant Ensemble Learner (REVEL) score of 0.377) could be not applied. Resultantly, p.R164W could be classified as ‘likely pathogenic’ according to the combining criteria to classify variants (Table 1) [18]. As for p.Arg557Glyfs*13, this variant could be classified as ‘pathogenic’ according to the combining criteria to classify variants (1 very strong (PVS1), 1 moderate (PM2), and 1 supporting evidence (PP1) [18]. As for p.Gly509Glu, this variant could be classified as variants classified as uncertain significance (VUS) according to the combining criteria to classify variants (1 moderate (PM2)) [18].

Collectively, this recurring novel missense variant, p.Arg164Trp of *PDZD7* was determined as the ‘likely pathogenic’. The remaining two novel variants, p.Gly509Glu and p.Arg557Glyfs*13 were also classified as the ‘VUS’ and ‘pathogenic’, respectively.

### 2.3. STR Marker Genotyping Data Analysis

To determine whether this variant (p.Arg164Trp) that was detected in our four unrelated families was derived from a common founder allele, we compared the haplotypes of available family members from all four families segregating p.Arg164Trp. The sequence of the microsatellite markers was as follows: D10S185, D10S520, D10S192, PDZD7, D10S1697, D10S1268, and D10S597. Reconstruction of the haplotypes, involving six STR markers from all five affected subjects (SB394–761, 762, SB297–597, SH246–577, and SB78–137) of four families, was accomplished based on the information from parents and affected subjects (Figure 4).

None of a single common haplotype shared by all four probands with p.Arg164Trp was identified by STR genotyping based on the six STR markers. Rather, eight different haplotypes for the six STR markers were observed. Specifically, SB394–761 (homozygote) with rigorously clarified PDZD7 haplotypes had two different haplotypes for p.Arg164Trp.

When three consecutive STR genotyping markers (D10S192, D10S1697, D10S1268) flanking *PDZD7* were considered for haplotype reconstruction, two unrelated probands (SB394–761 and SB78–137) carried an identical haplotype, and none of the controls (n = 81) presented this haplotype, showing a statistically significant association between p.Arg164Trp and this single haplotype, consisting of three consecutive STR genotyping markers (D10S192, D10S1697, D10S1268) (2/8 vs. 0/81, *p* = 0.007 by Fisher’s exact test) (Figure 4). 

### 2.4. A Systematic Review of Auditory Phenotype Due to Biallelic PDZD7 Variants

Here, a systematic review concerning the auditory phenotypes of patients with biallelic *PDZD7* variants is shown in Table 2.

## 3. Discussion

To the best of our knowledge, this is the first report to identify novel pathogenic variants of *PDZD7* from multiple non-consanguineous Korean families, segregating moderate-to-severe SNHL in an autosomal-recessive manner. In this study, two families demonstrated a homozygous missense variant (c.490C > T: p.Arg164Trp), whereas the remaining two families carried the p.Arg164Trp variant as one of the two variants as compound heterozygosity: one family with a compound heterozygous frameshift variant (c.1669delC:p.A557fs) with a missense variant (c.490C>T:p.Arg164Trp) and the other with two missense variants (c.1526G>A:p.G509E and c.490C>T:p.Arg164Trp). Conservation and pathogenicity-prediction analyses indicated that the pathogenic variants of *PDZD7* were responsible for ARNSHL. Notably, a recurring novel missense variant (p.Arg164Trp) in the PDZ1 domain was found in 70% of the alleles of five affected subjects. According to PDZD7 domains map (Figure 5), p.Arg164Trp resided in the distal end of PDZ1 domain and exon 4. Of five missense variants before our study, three variants, including p.Gly103Arg (PDZ1 domain), p.Gly228Arg (PDZ2 domain), and p.Met265Arg (PDZ2 domain), were predicted to be pathogenic since the alteration of intermolecular PDZ interactions may perturb the maintenance of USH2 interactome. Conversely, the remaining two missense variants, including p.Met64Ile and p.Arg66Leu, lie outside functional domains, but these variants also predicted to have a deleterious result due to loss of H-bond, possibly destabilizing the amino acid side chain.

Although *PDZD7* variants represent a rare cause of ARSNHL, p.Arg164Trp of this gene appears to be an important contributor causing moderate-to-severe ARNSHL at Koreans, considering the high prevalence of p.Arg164Trp based on information in the KRGDB (0.003205/11). That is, the recurring p.Arg164Trp missense variant can identify at least one allele in every 150 normal hearing people in Koreans. *SLC26A4* and *GJB2* variants were identified as the most prevalent molecular etiologies in prelingual ARSNHL, accounting for 18% and 17.2% of all cases studied, respectively [19]. Specifically, the frequency of p.H723R—the most common variant of *SLC26A4* causing EVAS—was reported as being approximately one allele in 70 control subjects in Korea [20]. Thus, p.Arg164Trp may theoretically occur at a rate of approximately 50% of EVA subjects carrying p.H723R of SLC26A4 variant at least in Koreans, which potentially affects clinical practice by highlighting the need for incorporating p.Arg164Trp into the molecular genetic testing as a cost-effective screening marker, especially in subjects with moderate-to-severe ARNSHL. Our study suggested that p.Arg164Trp variant could potentially exert a common founder effect and arise from diverse haplotypes as a mutational hot spot, as documented by p.Arg164Trp-linked STR genotyping. Based on our STR marker genotyping results where we observed the p.Arg164Trp was statistically significantly associated with a certain haplotype, consisting of three hypervariable informative STR markers (D10S192, D10S1697, D10S1268) flanking PDZD7 and higher prevalence in Korean MAF (minor allele frequency) population database (KRGDB) than in global population database (ExAC and GnomAD Exome) by at least one order of magnitude, we reasoned that this variant could exert a founder effect at least in Koreans. However, the effect does not seem to be predominant. The presence of short stretches of common haplotype shared by only two unrelated probands might indicate that this allele could be a very old founder allele that went through numerous recombination across generations, and therefore, got to retain a very narrow common haplotype. Indeed, STR genotyping, based on six STR markers, revealed various p.Arg164Trp-linked haplotypes shared by all of the affected subjects. In particular, two different haplotypes, related to p.Arg164Trp, were found in one carrier (SB394–761) with a homozygous missense variant, suggesting that p.Arg164Trp might also be a mutational hot spot. Identifying a mutational hot spot can be important for large-scale screening for ARSNHL because a mutational hot spot can be frequently encountered, regardless of the specific ethnic background [15]. Besides, the discovery of a mutational hot spot via STR genotyping might provide insights into the complete phenotypic spectrum and the possible mechanism that causes ARNSHL. Therefore, we conclude that this variant (p.Arg164Trp) could exert a founder effect as well as serving a mutational hotspot. 

The presence of biallelic *PDZD7* variants was highly associated with ARNSHL, considering that all subjects carrying the genotype, including our affected subjects, demonstrated ARNSHL without definite syndromic USH features at least at the time of diagnosis. Although the previous study demonstrated that heterozygosity for a truncating PDZD7 variant could act as a modifier of retinal disease and a contributor to digenic USH [21], a new perspective on the genotype–phenotype correlation that reflects the diversity of phenotype depending on the number of involved alleles (bi-allele vs. mono-allele) requires further investigation. However, given that RP usually develops in the second decade of an individual’s life [21], a follow-up electroretinogram to completely exclude USH2 is needed at least when the subjects are 10 years old. Based on a literature review regarding the auditory phenotypes of patients with biallelic *PDZD7* variants (Table 2), most affected subjects (seven out of nine) in previous studies demonstrated symmetric moderate-to-severe SNHL with residual hearing at low frequencies [10,11,12,13,14]. In a study by Booth et al., the preservation of low-frequency hearing was maintained even in adults over the age of 30 years [10]. In a recent study, it was also suggested that the *PDZD7* variation might lead to milder progressive hearing loss [11], although, this assumption was limited by inter-rater comparisons among siblings of different ages, requiring longitudinal follow-up evaluations. Overall, the auditory phenotype in subjects with biallelic *PDZD7* variants appears to be a coherent phenotype, characterized by moderate-to-severe SNHL with residual hearing at low frequencies and with mild progression. Detection and preservation of low-frequency hearing was a key consideration for initiating auditory rehabilitation [22,23]. Thus, *PDZD7* can serve as an appropriate candidate gene for personalized and customized auditory rehabilitation. In this study, one of our probands (SB394–761) recently underwent cochlear implantation due to delayed language development, compared with control subjects at the same age. A soft cochlear implant surgery, aimed at preserving low-frequency hearing, was performed on this subject based on a possible genotype-auditory phenotype correlation with *PDZD7*.

Given that the variable structural and functional study results obtained from analyzing the loss of individual *USH2* genes were correlated with the severity of ankle-link complex disruption [24], the extent of disorganization of the ankle link complex and resultant auditory phenotype can be expected to be similar among subjects with biallelic *PDZD7* variants. However, there were two exceptional subjects who manifested more severe auditory phenotype without any residual hearing in low frequencies among carriers of biallelic *PDZD7* variant in the literature (Table 2) [12]: the one with a homozygous nonsense variant of *PDZD7* (p.Q526X) and the other carrying both a homozygous *CIB2* variant (c. 223G > A) and a homozygous *PDZD7* variant (p.G228R) [12]. CIB2 variants, which are responsible for DFNB48 [25,26], are likely to be related to the more severe auditory phenotype than DFNB57 because this CIB2 variant is predicted to alter protein conformation and negatively affect calcium binding [10]. Therefore, Booth et al. noted that this subject had severe SNHL, which is identical to DFNB48 deafness, due to the existence of a homozygous variant in CIB2 [6]. 

The observation that pathogenic biallelic variations in human *PDZD7* have been associated with ARNSHL in the literature was in agreement with a previous study employing *PDZD7*-knockout mice. *PDZD7* deletion led to different spatial effect in the USH2-complex organization between hair cells and photoreceptors, indicating that PDZD7 was essential for the development of USH2 complex at ankle links, exclusively in cochlear hair cells but not photoreceptors [7]. Given the scaffolding ability of PDZD7 that facilitates multi-complex interactions [8,27], it remains uncertain why biallelic *PDZD7* variants manifest only hearing loss. Ebermann et al. demonstrated that PDZD7 showed dynamic interactions and differential binding affinities with members of the USH2 interactome, indicating that PDZD7 preferentially interacted with ADGRV1 rather than with USH2A [8]. The absence of other USH features, such as RP, might be attributed to the possible compensation by PDZD7 protein homologs, such as harmonin and whirlin (WHRN). Indeed, the WHRN and PDZD7 proteins are paralogs, with a sequence similarity exceeding 50% [28]. Similarly, an in vitro study suggested that homologous proteins might compensate for reducing the differential symptoms attributable to *PDZD7* variants. For example, CADM1 is a PDZD7-binding protein that is highly expressed in the inner ear in mice, but *Cadm1* knockdown did not elicit an abnormal hearing threshold, suggesting that its homologs might compensate for CADM1 loss in the inner ear [27]. Further studies are warranted to clarify phenotypic differences occurring in the context of the complex interplay between members of the USH2 interactome.

## 4. Materials and Methods

### 4.1. Subjects

All procedures in this study were approved by the Institutional Review Boards of Seoul National University Bundang Hospital (IRB-B-1007–105–402). Written informed consent was obtained from both the affected and unaffected individuals included in the study. For children, written informed consent was obtained from their parents or guardians on their behalf.

Four families (SB94, SB297, SH246, and SB78), showing moderate-to-severe SNHL in an autosomal-recessive fashion, were enrolled in this study. Subjects underwent comprehensive phenotypic evaluations, including medical history interviews, physical examinations, imaging, and audiological assessments. Through a meticulous review of their medical histories, we found that none of the affected subjects had developed developmental delays, brain abnormalities, or cytomegalovirus infections. For imaging in all affected subjects, TBCT or IAC MRI analyses were conducted to identify phenotypic markers related to hearing loss, such as EVAS and incomplete partitions. Also, ophthalmologic and vestibular evaluations were performed in the affected subjects, where possible.

### 4.2. Audiological Evaluations

ABRT, ASSR, and DPOAE analyses were performed with all affected subjects. Furthermore, age-appropriate behavior tests, such as play audiometry and PTA (pure tone audiogram), were performed to evaluate the hearing level before and after hearing aid application.

To review the audiological characteristics of previous cases carrying *PDZD7* variants, we attempted to document the following four properties of hearing based on PTA: severity, residual hearing at different frequencies, configuration, and symmetry. The hearing threshold was calculated by averaging the thresholds at 0.5, 1, 2, and 4 kHz, and the severity of hearing loss was classified as being mild (20–40 dB), moderate (41–55 dB), moderately severe (56–70 dB), severe (71–90 dB), and profound (>90 dB). The range of hearing loss, depending on the PTA results in available subjects, was described in the following manner: low frequency, 250–500 Hz; mid-frequency, 1–2 kHz; high frequency, 4–8 kHz. The audiogram configurations were classified as down-sloping, rising, or flat across various frequencies. Specifically, the down-sloping auditory configuration was defined by increasing thresholds from 0.25 to 8 kHz, and the difference between the thresholds at these frequencies was >20 dB hearing loss [29]. Furthermore, a difference between the left and right ear air-condition threshold of >20 dB with at least two frequencies out of 0.5, 1, and 2 kHz was indicated as being asymmetric. 

### 4.3. Molecular-Genetic Testing

A molecular-genetic diagnosis was made for four families (SB94, SB297, SH246, and SB78), as described previously. Paternal DNA samples from both families were available to investigate the segregation. For molecular-genetic testing, Exome Sequencing was employed on four proband samples. The obtained reads were aligned with the University of California, Santa Cruz (UCSC) hg19 reference genome browser (https://genome.ucsc.edu/), which is an interactive website offering access to genome sequence data from various vertebrate and invertebrate species (including major model organisms) that is integrated with a large collection of aligned annotations, with narrowed-down variants. Bioinformatics analysis was then performed to select candidate variants in autosomal-recessive genes related to non-syndromic SNHL, as follows.

In brief, the raw data generated by Exome sequencing were mapped onto the UCSC hg19 reference genome, and rare single-nucleotide variations (SNVs), indels, or splice-site variants were selected via the following filtering process: (i) a basic filtering step, which entailed excluding synonymous SNVs and selecting SNVs with a quality score of greater than 30 and a read depth of more than 20; (ii) compatibility with the autosomal-recessive inheritance pattern; (iii) confirmation of the presence of variants by Sanger sequencing; (iv) a segregation study and/or control study against the KRGDB consisting of 1722 Korean individuals (3444 alleles) (http://152.99.75.168/KRGDB/menuPages/firstInfo.jsp); (v) compatibility with clinical features. SNVs with MAFs ≤0.005 were chosen. Global MAFs were verified using several databases, including ExAC, 1000 Genomes, and ESP.

We constructed approximately 200 genes list related to hearing loss or deafness phenotype through hereditary hearing loss (https://hereditaryhearingloss.org/), deafness variation database (http://deafnessvariationdatabase.org/), OMIM (https://omim.org/). We have added this in the methods. To predict the pathogenic potential of each detected variant, and compatibility with inheritance and audiogram patterns, an in silico study using SIFT (http://sift.jcvi.org/), and PolyPhen2 (http://genetics.bwh.harvard.edu/pph2/) were performed to predict the missense variants. Additionally, to estimate the evolutionary conservation of the amino acid sequences, we referred to the GERP + + score from the UCSC Genome Browser (http://genome.ucsc.edu/).

### 4.4. Predicting Pathogenic Potentials Based on the MAFs of Candidate PDZD7 Alleles

To study the potential pathogenicity of specific PDZD7 variants, MAFs were verified using the ExAC, GnomAD, and KRGDB, which provides a comprehensive map of the Korean genomic variants for future disease-association and population-genetics studies. The MAFs of the alleles, including p.Arg164Trp and three other variants (consisting of homozygous, frameshift, and in trans variants), were calculated and compared with the allele frequencies among Korean patients with prelingual ARSNHL. A schematic representation of the variant-filtering strategy employed with the Exome sequencing data is illustrated in Figure 2. 

### 4.5. STR Marker Genotyping

The haplotypes of the *PDZD7* p.Arg164Trp alleles in Korean subjects were determined to identify whether the mutant allele is a common founder variant or a mutational hot spot. For this purpose, we selected six STR markers flanking the *PDZD7* gene, which included three markers (D10S185, D10S192, and D10S597) previously reported to be hypervariable and informative. Primers for the six STR markers flanking *PDZD7* were generated based on the UCSC Genome Browser, as previously described. The sequence of the microsatellite markers was as follows: D10S185, D10S520, D10S192, PDZD7, D10S1697, D10S1268, and D10S597.

Loci were amplified using AmpliTaq DNA polymerase (Invitrogen Life Technologies, Carlsbad, CA, USA) in a Perkin Elmer 9700 thermal cycler (Perkin Elmer, Waltham, MA, USA). The genotypes of these loci on 10 chromosomes were defined using the AmpliTaq Gold (Applied Biosystems, Waltham, MA, USA) touchdown protocol under the following conditions: 10 min at 95 °C; 10 cycles of 30 s at 94 °C, 30 s at 65 °C, and 30 s at 72 °C; a further 20 cycles under the same conditions but with the annealing temperature lowered to 60 °C for 5 cycles and then 55 °C for 15 cycles, and a final extension for 10 min at 72 °C. STR marker genotyping was performed, as described previously [22]. We compared the variant-linked haplotypes of the available family members in the three families with the haplotypes of 81 adult Korean subjects with documented normal hearing thresholds.

## 5. Conclusions

In conclusion, we report here for the first time that *PDZD7* can act as a disease-causing gene in Korean families, as evidenced by the segregation of recessive inherited moderate-to-severe hearing loss. All affected subjects demonstrated that the presence of biallelic *PDZD7* variants is responsible for ARNSHL. Notably, p.Arg164Trp was frequently identified, and this variant was characterized as both a mutational hot spot and potentially having a founder effect by STR marker genotyping. Our results suggest that screening for p.Arg164Trp can narrow down the candidate population for subjects with moderate-to-severe ARNSHL for more efficient genetic diagnosis. Also, *PDZD7* might serve as an appropriate candidate gene for personalized and customized auditory rehabilitation, based on a possible genotype–phenotype correlation observed in this study.

## Figures and Tables

**Figure 1 ijms-20-04174-f001:**
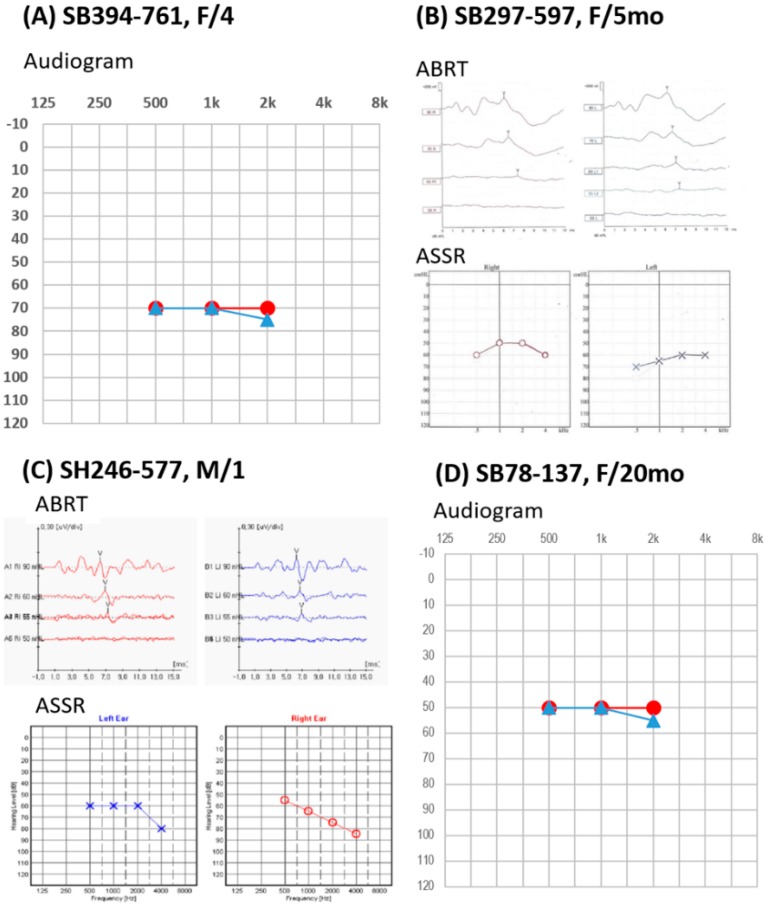
Audiological results of four unrelated families segregating mutations in PDZD7. The horizontal axis shows tone frequency (Hz); the vertical axis gives the hearing level (dB). Symbols “●” and “▲” denote air conduction pure-tone thresholds at different frequencies in the right (red color) and left ear (blue color). (**A**) For SB394-761 (F/4), non-syndromic severe sensorineural hearing loss (SNHL) was obtained by play audiometry with air conduction at frequency 500 Hz, 1k Hz, and 2k Hz. (**B**,**C**) For SB297-597 (F/5mo) and SH246-577 (M/1), the response to 55 dB click sounds on both ears and symmetrical moderate-to-severe non-syndromic SNHL were observed in the auditory brainstem response threshold (ABRT) and auditory steady-state response (ASSR), respectively. (**D**) For SB78-137 (F/20mo), non-syndromic SNHL was obtained by play audiometry with air conduction at frequency 500 Hz, 1 kHz, and 2 kHz.

**Figure 2 ijms-20-04174-f002:**
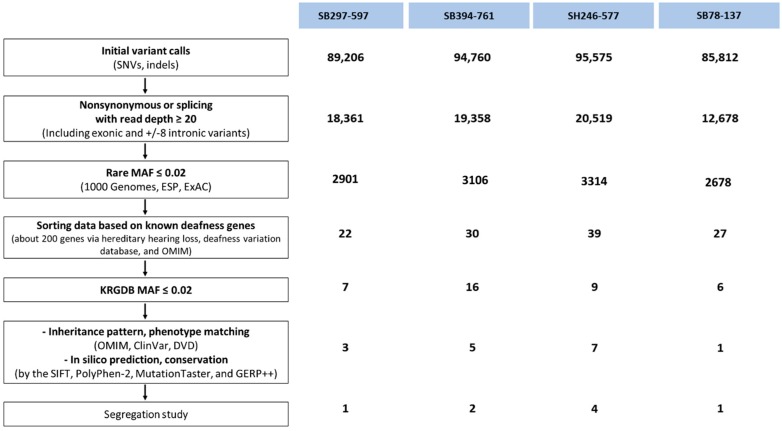
Variant filtering strategy of Exome sequencing data and the number of candidate variants in four unrelated families. Single-nucleotide variations, SNVs; MAF, minor allele frequency; Exome Aggregation Consortium databases; ESP, Exome Sequencing Project; ExAC, Exome Aggregation Consortium databases; KRGDB, Korean reference genomic database; OMIM, Online Mendelian Inheritance in Man; DVD, Deafness variation database; SIFT, Sorting Intolerant from Tolerant; GERP, Genomic Evolutionary Rate Profiling.

**Figure 3 ijms-20-04174-f003:**
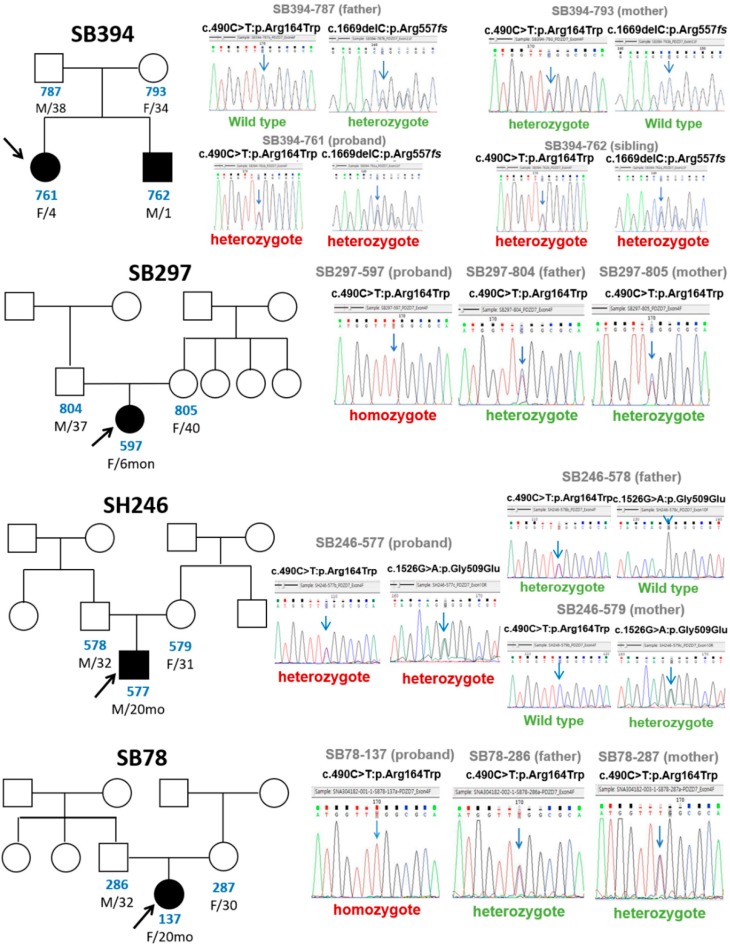
Pedigrees and sanger sequence chromatograms for families segregating mutations in PDZD7. Males are denoted with squares and females with circles. Solid shapes with an arrow are individuals with reported hearing loss (proband). Sanger sequencing demonstrates each mutation in the chromatogram from affected individuals and heterozygous variant from unaffected parents.

**Figure 4 ijms-20-04174-f004:**
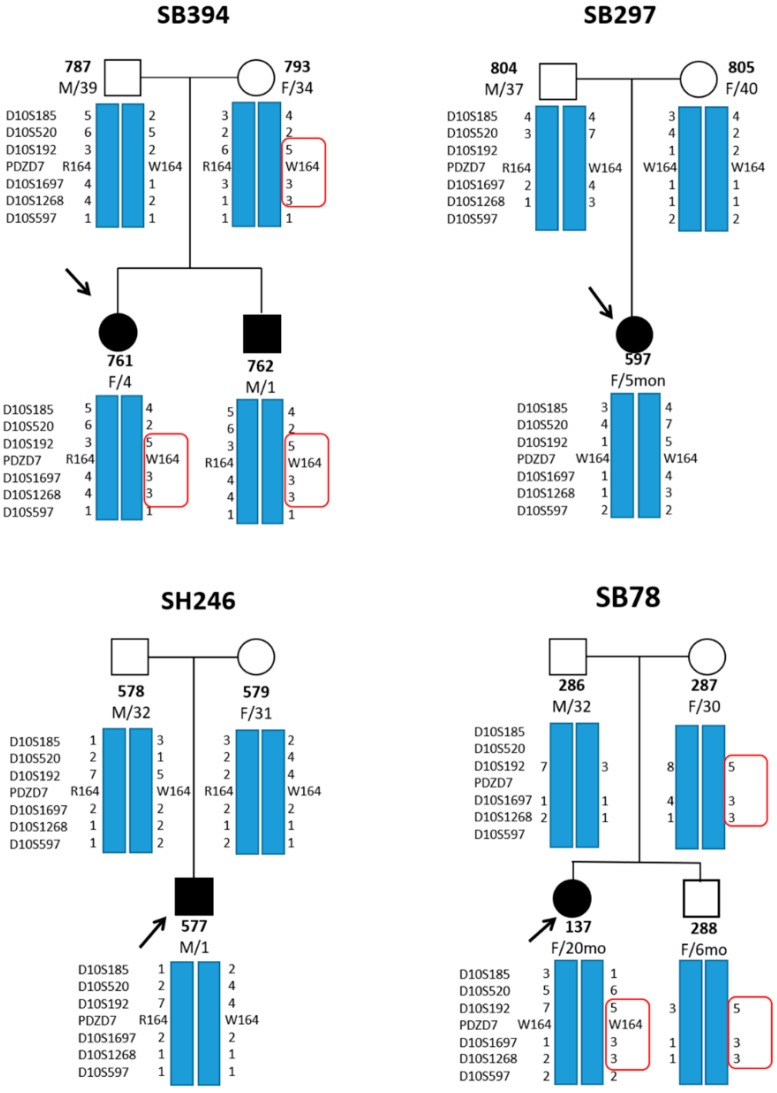
Pedigrees, showing p.Arg164Trp-linked haplotypes in the DFNB57 locus from four unrelated Korean families. The sequence of the microsatellite markers was as follows: D10S185, D10S520, D10S192, PDZD7, D10S1697, D10S1268, and D10S597. A red box indicates short stretches of common haplotype consisting of only three consecutive STR genotyping markers (D10S192, D10S1697, and D10S1268). Two unrelated probands (SB394–761 and SB78–137) carried an identical haplotype, and none of the controls presented this haplotype. Solid shapes with arrows denote the probands.

**Figure 5 ijms-20-04174-f005:**
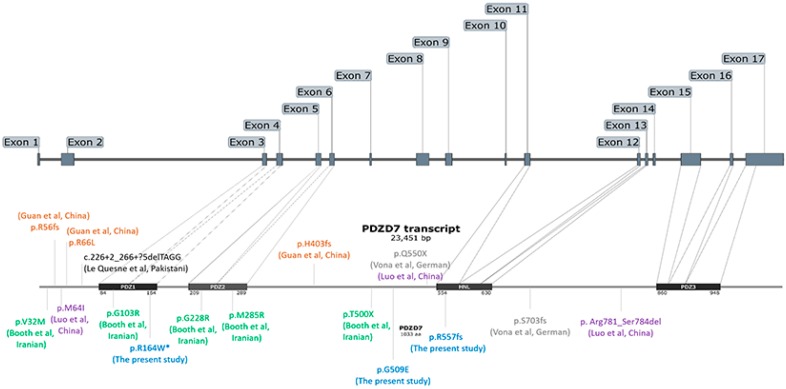
Schematic diagram of the PDZD7 domains map. Previously reported and our novel variants of PDZD7 are presented. The recurring novel missense variant (p.Arg164Trp, asterisk) in the present study resides in the exon 4 encoding PDZ1 domain.

**Table 1 ijms-20-04174-t001:** Pathogenic variants of PDZD7 and its pathogenicity from four unrelated Korean families.

Gene.	Family ID.	HGVS Nucleotide and Protein Change [NM_001195263.1] :[NP_001182192.1]	Zygosity	Prediction Analysis of Variants			Nucleotide Conservation Prediction		MAF		Reported/ Novel	ACMG-AMP [14,15]	
				Mutation Taster ^a^	PP-2 ^b^	SIFT ^c^	PhyloP ^d^	GERP + + ^e^	GMAF ^f^	KRGDB ^h^ (1722 individuals)		Criteria	Classification
*PDZD7*	SB 297	c.490C>T :p.Arg164Trp	Homo	DC	1(D)	0.014(D)	−0.476	−1.62	0.00005/6 (ExAC) 0.00004775/12 (GnomAD exome)	0.003205 (11/3444 alleles)	Novel (this study)	PM2, PM3_Strong PP1	‘likely pathogenic’
	SB 394	c.490C>T :p.Arg164Trp	Comphet	DC	1(D)	0.014(D)	−0.476	−1.62	0.00005/6 (ExAC) 0.00004775/12 (GnomAD exomes)	0.003205 (11/3444 alleles)	Novel (this study)	PM2,PM3_Strong PP1	‘likely pathogenic’
		c.1669del :p.Arg557Glyfs*13		DC	ND	ND	1.751	4	ND (ExAC, GnomAD)	ND	Novel (this study)	PVS1,PM2, PP1	‘pathogenic’
	SH 246	c.490C>T :p.Arg164Trp	Comphet	DC	1(D)	0.008(D)	−0.476	−1.62	0.00005/6 (ExAC) 0.00004775/12 (GnomAD_exomes)	0.003205 (11/3444 alleles)	Novel (this study)	PM2, PM3_Strong PP1	‘likely pathogenic’
		c.1526G>A :p.Gly509Glu		DC	0.123(B)	0.057(T)	1.863	3.61	0.00005/6 (ExAC) 0.00004704/7 (GnomAD_exomes)	0.003205 (11/3444 alleles)	Novel (this study)	PM2,	‘VUS’
	SB78	c.490C>T :p.Arg164Trp	Homo	DC	1.0(D)	0.008(D)	−0.476	−1.62	0.00005/6 (ExAC) 0.00004775/12 (GnomAD_exomes)	0.003205 (11/3444 alleles)	Novel (this study)	PM2,PM3_Strong PP1	‘likely pathogenic’

HGVS, human genome variation socitey; Homo, homozygosity; Comp Het, compound heterozygosity; DC, disease causing; D, deleterious; T, tolerated; B, benign; ND, not determined; VUS, variants classified as uncertain significance ^a^ Mutation taster (http://www.mutationtaster.org/). ^b^ PolyPhen-2 (PP2) prediction score (HumanVar), ranges from 0 to 1 (0 = benign, 1 = probably damaging [http://genetics.bwh.harvard.edu/pph2/]). ^c^ Sorting Intolerant from Tolerant (SIFT; http://sift.jcvi.org/). ^d^ phylogenetic *p*-values (PhyloP), ranges from –20 to 10. ^e^ Genomic Evolutionary Rate Profiling (GERP + +; http://genome.ucsc.edu/). ^f^ Exome Aggregation Consortium databases (ExAC), and Genome Aggregation Database (GnomAD). ^h^ Korean reference genomic database (KRGDB; http://coda.nih.go.kr/coda/KRGDB/index.jsp).

**Table 2 ijms-20-04174-t002:** A systematic review of auditory profiles in subjects with PDZD7 biallelic variants.

Subject	Origin	Age/Sex	Biallelic Varinats							Auditory Phenotype					Reference
			Gene	Type	Genotype	HGVS.c	HGVS.p	Location	Domain	Onset	Severity	Configuration	Symmetry	Progression	
1	China	11yr/M	PDZD7	Frameshift Frameshift	Comp Het	c.166_167insC* c.1207delC	p.R56fs p.H403fs	Exon2 Exon8		prelingual	moderate tosevere	down-sloping	yes	(-)	Guan et al. [11]
2	China	7yr/M	PDZD7	Missense Nonsense	Comp Het	c.192G>A c.1648C>T	p.M64I p.Q550X	Exon2 Exon10		prelingual	mild to moderate	down-sloping	NA	(-)	Luo et al. [9]
				Frameshift		c.2341_2352del CGCAGCCGCAGC	p.R781_ S784del	Exon14							
3	Pakistani	N/A	PDZD7	Frameshift	Homo	c.226 + 2_266 + 5 5delTAG	NA	Intron2		congenital	moderate	NA	NA	(-)	Le Quesne Stabej et al. [13]
4	China	10yr/F	PDZD7	Missense	Homo	c.197G>T	p.R66L	Exon2		prelingual	moderate to severe	down-sloping	yes	(-)	Guan et al. [11]
5	Iranian	36yr/F	PDZD7	Missense	Homo	c.307G>C	p.G103R	Exon3	PDZ1	prelingual	moderate to severe	down-sloping	yes	NA	Booth et al. [10].
6	South Korea	6mo/M	PDZD7	Missense	Homo	c.490C>T	p.R164W	Exon4	PDZ1	prelingual	moderate to severe	NA	yes	(-)	Present study
7	South Korea	4yr/F	PDZD7	Missense Frameshift	Comp Het	c.490C>T c.1669delC	p.R164W p.R557fs	Exon4 Exon11	PDZ1	prelingual	Severe	NA	yes	(-)	Present study
8	South Korea	20mo/M	PDZD7	Missense Missense	Comp Het	c.490C>T c.1526G>A	p.R164W p.G509E	Exon4 Exon10	PDZ1	prelingual	moderate to severe	NA	yes	(-)	Present study
9	South Korea	20mo/F	PDZD7	Missense	Homo	c.490C>T	p.R164W	Exon4	PDZ1	prelingual	Moderate	NA	yes	NA	Present study
10	Iranian	33yr/F	CIB2 PDZD7	Missense	Homo Homo	c.94G>A	p.V32M	Exon5	EF	prelingual	severe	flat	yes	NA	Booth et al. [12]
				Missense		c.682G>A	p.G228R								
11	Iranian	38yr/F	PDZD7	Missense Nonsese	Comp Het	c.854T>G c.1500C>A	p.M285R p.T500X	Exon6 Exon9	PDZ2	prelingual	moderate to severe	down-sloping	yes	NA	Booth et al. [10]
12	Iranian	23yr/F	PDZD7	Nonsense	Homo	c.1576C>T	p.Q526X	Exon9		prelingual	severe	flat	yes	NA	Booth et al. [10]
13	German	10yr/F	PDZD7	Nonsense	Comp Het	c.1648C>T	p.Q550X	Exon10		prelingual	moderate to severe	down-sloping	yes	(-)	Vona et al. [12].
				Frameshift		c.2107del	p.S703fs	Exon14							
14	German	15yr/M	PDZD7	Reciprocal translocation	Homo	t(10;11)	NA	Intron10		prelingual	moderate to severe	down-sloping	yes	HFHL	Vona et al. [12]. Schneider et al. [14]

Abbreviation: M, male; F, female; yr, year; mo, month; HFHL, high-frequency hearing loss; Homo, homozygosity; Comp Het, compound heterozygosity; NA, not available.

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
