# Peer review of "Identification of a Potential Founder Effect of a Novel PDZD7 Variant Involved in Moderate-to-Severe Sensorineural Hearing Loss in Koreans"

_ijms, 2019, doi:10.3390/ijms20174174_

Round 1

Reviewer 1 Report

Lee and colleagues describe four families with autosomal recessive non-syndromic moderate-severe hearing loss. Using exome sequencing they identified bi-allelic variants in the PDZD7 gene. One of which, p.R164W, was recurrent and determined to be a founder mutation.

Overall, the manuscript is well written, logical, used appropriate methods for the investigation and the conclusions are supported by the results.

There are a few areas that could improve the quality of the manuscript.

General comments:

The phrase “it was recently shown…” is used too much. This phrase should only be used in the case that the manuscript being referenced is < 1 year old.

There is too much discussion in the results section. Please only state what the results were. Save all interpretation and comparison of the data for the discussion.  

Line 44: Missing citation after “DFNB57”.

Line 55: “less than 10 subjects” is not accurate. In the original describing PDZD7 as the cause of DFNB57 HL, the authors present 12 subjects. To date only 10 variants have been linked to DFNB57-HL. There is a difference between patients and number of variants.

For the results please rearrange so that the clinical and patient information is presented first followed by: Variant identification, segregation, STR markers and variant classification.

Line 66: The title of this section should be “Variant Identification and Classification”

Line 69: WES should read; “Exome Sequencing”. Based on ACMG recommendations, there term “Whole Exome Sequencing” should be referred to as “Exome Sequencing”. Please change throughout the manuscript.

The authors should use gnomAD frequencies whenever possible. Please change this in the text and tables.

Table 1:

Add genomic positions. Separate ACMG-ACMG Criteria into 2 columns. One column should be named “ACMG Criteria” the next column should be name “ACMG Classification”. “Reported/Novel” should be read “Ref” and the value in each column should be “This study”.

Line 117-118: Should be removed.

The authors need to be care not to discuss their results in the results section. Lines 142-145 should be in the discussion section.

Figure 4:

Add the Markers to the left hand side for each generation. It is unclear which patients carry the p.R164W based on this figure and what their genotype is for this variant. Please alter the figure to clarify this.

Figure 5:

In its current state this is very hard to understand. Please make each patient’s data its own panel. For example: The audiogram for SB394, should be “A” and would be referenced in the text as “Figure 5A”. Remove the word “Play” Remove the symbols and write “filled circles” and “filled triangles” Add age the data was collected after the patient ID. For example: “SB297 (6mo)” It is also unclear which sibling from SB394 audiogram is being presented

Figure 3 should become the final figure.

In the discussion the authors should talk about where these variants occur in the protein compared to other missense variants.

Table 2:

Add genomic position “Exon” should be “Location” “Effect” should be “Type” Reorder variants to be in ascending cDNA position.

Line 136: The conclusion the authors come to about the p.R164W is confusing. The authors suggest this is a founder mutation, however the haplotypes carrying this allele are different. The data suggests this variant arose on several different haplotypes. Are all the families from the same regions in Korea? While not required, more markers would help further refine and resolve these haplotypes. The authors need to revise this section and clarify. Also please make sure there is consistency about the conclusion of this variant, hot-spot or founder mutation, as the discussion leaves it unclear (lines 230-246).

Line 220-229: The authors are extrapolating their results too much. They are comparing their results to studies (and combined studies) which over the years have screen thousands if not tens-of-thousands of patients for these variants. This study has only screen five affected individuals. Please tone down this part about this allele’s contribution to HL in the Korean population.

Lines 242-245: This sentence is confusing. How do these results “juxtapose” previous results to other genes?

Lines 246-250: There is not proposed genotype-phenotype correlation for DFNB57. The role of PDZD7 as a modifier for USH or involved in digenic HL is weak and the rational is missing. Please change this.

Lines 269-289:

Booth et al explained the more severe phenotype due to the mutation in CIB2. DFNB48 deafness is more severe than DFNB57, therefore the patient has the phenotype which is identical to DFNB48 deafness. Environmental causes are not the driver for the phenotypic difference. Mutations in CIB2 do not cause USH1J (see PMID: 29112224 and PMID: 29084757), remove this from the discussion. There is no evidence that CIB2 or PDZD7 interact. Please remove this part from the discussion. Line 280: DFNB56 should be DFNB57. The conclusion the authors reached here are wrong about what is discussed in Booth et al.

Please add a legend for the supplemental figure and ensure its referenced in the text.

Author Response

Dear reviewer,

Best regard.

Reviewer 2 Report

Summary:

Lee et al performed exome sequencing on four Korean families with at least one proband with moderate-severe nonsyndromic hearing loss and identified variants in PDZD7. One variant, p.R164W, was common across all of the cases and was identified in the compound heterozygous or homozygous state. To determine if this variant was a founder variant, STR analysis was performed and two families shared STR markers. The authors claim that this variant is a founder variant and that variants in PDZD7 are associated with NSHL in the Korean population.

Minor Points:

-There are some typos in the manuscript. To point out some that I found:

line 131 “flaking” should be “flanking”

line 136 “Fischer’s Exact Test” should be “Fisher’s Exact Test”

line 212

-The most up-to-date gene name for GPR98 is ADGRV1. Please change accordingly or at least acknowledge that they are the same gene.

-Please list the transcripts used for all of the HGVS nomenclature (in the methods is fine)

-Figure 2 legend: The authors mention “Generation and individual identification are noted by Roman numerals”. Please fix this. There are no Roman Numerals in the pedigrees

Major Points:

-Exome filtration for a paper submitted this recently should really use ExAC or gnomAD for minor allele frequency filtration. In this case, your filtration usage of variants that are rare in 1000 Genomes/ESP is a more conservative method, so it is still acceptable. Comment on why newer population databases were not used for filtration analysis.

-In figure 1, what do the authors mean by “known deafness genes”? Please elaborate on where this list comes from in the methods.

-In figure 2, it is important to know if the compound heterozygous individuals have been phased. This is more important than demonstrating that the homozygous observations are present in each parent. Where the parents tested/genotyped for families SB394 and SH246? Please include parental sequencing information that you have displayed on the pedigrees.

-ACMG code application for the variants is not appropriate:

In general, nonsyndromic hearing loss is too non-specific to apply PP4. There are many genes that can give this phenotype and PP4 is meant for much more of a specific (often syndromic phenotype).

PM1 is also loosely applied. No functional evidence is mentioned to define the 164 residue as a hot spot.

However, the authors (if the computational predictors do predict appropriate pathogenicity) could apply PP3.

PM3 could be upgraded for multiple case observations. See Oza et al (PMID: 30311386) for a suggestion of how to upgrade PM3 for hearing loss genes.

*It’s not clear with these updated codes if any of the missense variants would make it to likely pathogenic.

-The way that the MAF for the Korean database is displayed is confusing. I’m not sure what 0.003205/11 means

-For exome filtration analysis, there is a list of high frequency pathogenic variants that cause hearing loss that could easily be filtered out if MAF was used as a primary filter. Please see Oza et al (PMID: 30311386) for this list and make sure that none of the families had these variants, especially p.M34T and p.V37I in GJB2. Please comment in the methods.

-If this variant were a common founder variant as the authors propose, one would expect it to appear in the heterozygous state in unaffected controls, but it is absent. Please comment.

-PDZD7 has already been definitively associated with nonsyndromic hearing loss. Please see the ClinGen curation found here:

https://search.clinicalgenome.org/kb/gene-validity/8406

Author Response

Dear reviewer,

Best regard.

Round 2

Reviewer 2 Report

Thank you for addressing all of the comments and points I raised. I believe this manuscript has been considerably strengthened and is acceptable for publication. I don’t have any other comments or concerns.